# Testis-Specific Protein Y-Encoded (TSPY) Is Required for Male Early Embryo Development in *Bos taurus*

**DOI:** 10.3390/ijms24043349

**Published:** 2023-02-08

**Authors:** Na-Young Rho, Teresa Mogas, W. Allan King, Laura A. Favetta

**Affiliations:** 1Reproductive Health and Biotechnology Lab, Department of Biomedical Sciences, Ontario Veterinary College, University of Guelph, Guelph, ON N1G 2W1, Canada; 2Department of Medicine and Animal Surgery, Autonomous University of Barcelona, Cerdanyola del Vallés, 08193 Barcelona, Spain; 3Karyotekk Inc., Ontario Veterinary College, University of Guelph, Guelph, ON N1G 2W1, Canada

**Keywords:** copy number variation (CNV), sex-sorted semen, in vitro fertilization, male development, testis-specific protein Y-encoded (TSPY)

## Abstract

*TSPY* is a highly conserved multi-copy gene with copy number variation (CNV) among species, populations, individuals and within families. *TSPY* has been shown to be involved in male development and fertility. However, information on *TSPY* in embryonic preimplantation stages is lacking. This study aims to determine whether *TSPY* CNV plays a role in male early development. Using sex-sorted semen from three different bulls, male embryo groups referred to as 1Y, 2Y and 3Y, were produced by in vitro fertilization (IVF). Developmental competency was assessed by cleavage and blastocyst rates. Embryos at different developmental stages were analyzed for *TSPY* CN, mRNA and protein levels. Furthermore, *TSPY* RNA knockdown was performed and embryos were assessed as per above. Development competency was only significantly different at the blastocyst stage, with 3Y being the highest. *TSPY* CNV and transcripts were detected in the range of 20–75 CN for 1Y, 20–65 CN for 2Y and 20–150 CN for 3Y, with corresponding averages of 30.2 ± 2.5, 33.0 ± 2.4 and 82.3 ± 3.6 copies, respectively. *TSPY* transcripts exhibited an inverse logarithmic pattern, with 3Y showing significantly higher *TSPY*. TSPY proteins, detected only in blastocysts, were not significantly different among groups. *TSPY* knockdown resulted in a significant *TSPY* depletion (*p* < 0.05), with no development observed after the eight-cell stage in male embryos, suggesting that TSPY is required for male embryo development.

## 1. Introduction

In humans, 20–30% of infertility is male-related, and almost 50% of these causes are idiopathic [1]. Chromosome abnormalities, microdeletion of the azoospermia factor (AZF) regions of the Y chromosome, genetic polymorphism and association with specific sequence repeats have been shown to affect spermatogenesis and male fertility [2,3,4].

Copy number variation (CNV), structural variation of the genomic sequence by deletion, insertion and duplication resulting in different number of copies of a gene, is associated with genetic diversity among individuals and species [5,6]. Studies have shown that an increased or decreased number of gene copies results in abnormal recombination and meiosis, leading to germ cell loss which further results in infertility [7]. CNV in the male specific region of the Y chromosome (MSY) is a result of rearrangement and duplication, which are known to occur frequently as a consequence of intrachromosomal interactions of non-allelic homologous recombination (NAHR) that alter the amplification, deletion or gene conversion [8,9].

*Testis-specific protein Y-encoded* (*TSPY*) is a male specific gene that is located in the MSY [10,11], mostly on the ampliconic region of the Yp (DYZ5) near the centromere, but with evidence of also being on the Yq arm [12,13,14]. *TSPY* is a highly conserved multi-copy gene with CNV among species, populations, individuals and within families. Rats have only one functional *TSPY* copy [15], while mice have a single pseudogene [16]. Pigs have three copies but are not CN variant [17]. Humans have 30–60 [18] *TSPY* CN, and the largest range of variation can be observed in cattle with a range of 20–200 copies [19]. *TSPY* CN is associated with male specific functions. In humans, higher *TSPY* CN was present in infertile men and men with prostate cancer [18]. In bovines, bulls with higher *TSPY* CN showed higher fertility rates [20], whereas low *TSPY* CN was linked to bulls with inferior semen production [21]. *TSPY* CN shows temporal changes during aging both in vitro and in vivo [22], suggesting possible epigenetic recombination of *TSPY*. *TSPY* is expressed exclusively in the germ cells during embryogenesis and spermatogenesis and is present in both prenatal and adult testis [13,23]. *TSPY* shows transcript heterogeneity which codes over 290 amino acids and allows alternative splicing patterns that result in a variety of polymorphic proteins [11,13,24]. Importantly, TSPY protein is conserved within the SET/NAP domain that interacts with the Cyclin B-binding domain [13] and plays an important role in promoting differentiation and proliferation during spermatogenesis, especially during spermatonoia proliferation, self-renewal, and at spermatocyte differentiation at prophase I [25,26]. TSPY levels are also associated with gonadoblastoma, and the tumorigenic capability in gonad and prostate cancers [27,28,29]. Most TSPY studies have been conducted in adult and fetal tissues and cells, but not much is known about the earlier preimplantation stages of development, which are key for fertility [30], although *TSPY* expression has been shown at the bovine blastocyst stage. There is substantial evidence that copy number varies among brothers and changes with aging [22]. However, in early embryos, it has only been superficially examined [30,31]. Therefore, the aim of the present study is to investigate whether TSPY is present and expressed and to determine its function during bovine preimplantation male embryo development for use as a translational model for humans [32].

## 2. Results

### 2.1. Early Embryo Developmental Percentage

Cleavage and blastocyst rates from groups produced by sex-sorted semen from the three different bulls and the unsorted semen (control) from the conventional bull semen are shown in Table 1 and Figure 1A,B. No significant differences in cleavage rate between oocytes fertilized with control semen and Y-bearing sperm were detected (Figure 1A). However, the cleavage rate of oocytes fertilized with X-chromosome bearing sperm was significantly lower than the control and Y-chromosome bearing groups (*p* < 0.05). There were no significant differences among the female embryo groups nor among the male embryo groups. Specifically, the average cleavage rates in the combined X-bearing sperm group and the combined Y-bearing sperm group were 62.2 ± 3.8 and 78.6 ± 3.5, respectively. The overall blastocyst rate among the embryos derived from the Y-bearing sperm was almost double that of the X-bearing sperm group (29.0 ± 1.7 vs. 16.6 ± 2.0) (Figure 1B). The X- bearing groups were not significantly different from each other, but all were significantly lower than the control group and the Y-bearing groups. Two groups derived from the Y-bearing sperm were significantly (*p* < 0.05) lower than the control group and the 3Y-bearing group. Blastocysts from all groups were examined for morphology and total blastocyst cell number, which were normal and within the range of 110–130 blastomeres/blastocysts, respectively, with no significant differences among the groups.

### 2.2. TSPY Copy Number Variation (CNV)

*TSPY* CN was measured in 30 individually collected blastocysts from male and female groups (1X, 2X, 3X, 1Y, 2Y, 3Y). *TSPY* was not detected in the female group (1X, 2X, 3X) but was present in all the male groups (1Y, 2Y, 3Y), with CNV between the brother embryos shown in Figure 2A–C. *TSPY* CNV ranges were 20–75, 20–65 and 20–150 for 1Y, 2Y and 3Y groups, respectively, with average numbers of 30.2 ± 2.5, 33.0 ± 2.4 and 82.3 ± 3.6, respectively. The average number of the *TSPY* CN from the male group was significantly higher than the number in the 3Y group (*p* < 0.005) (Figure 2D).

### 2.3. TSPY mRNA Expression

There was no *TSPY* mRNA expression in female groups (1X, 2X and 3X) at any of the pre-implantation embryonic developmental stages. However, in male groups (1Y, 2Y and 3Y), *TSPY* mRNA levels were detected throughout the first week of embryonic development at the 2-, 4-, 8-cells and blastocyst stages, as shown in Figure 3A–C. The *TSPY* expression pattern in all male embryo groups resembled an inverse logarithmic pattern. Specifically, between the 2- and 4-cell stages, there were no significant differences in all male groups. However, an over 30-fold increase in *TSPY* mRNA expression was shown in the 8-cell stage compared with the 4-cell stage. From the 8-cell stage to the blastocyst stage, a 2-fold or higher increase was observed. Comparisons of *TSPY* mRNA levels between the three male groups at 2-, 4-, 8-cells and blastocysts are shown in Figure 3D. Overall, *TSPY* mRNA expression was significantly higher in the 3Y group at the 8-cell and the blastocyst stages.

### 2.4. TSPY Protein Localization and Quantification

TSPY protein was localized and expression was quantified in male (1Y, 2Y, 3Y) and female (1X, 2X, 3X) groups at different developmental stages (2-, 4-, 8-cells and blastocyst), using oocyte as a negative control. TSPY protein was not detected in any of the female groups and was only detected at the blastocyst stage in the male groups (Figure 4A). Specifically, TSPY was localized exclusively in the inner cell mass of male blastocysts. TSPY protein was quantified in male blastocysts, but the levels were not significantly different among the three bulls (Figure 4B).

### 2.5. TSPY Knock-Down

#### 2.5.1. Unsorted Sex Semen

Validation of the role of TSPY during embryo development was performed by *TSPY* knockdown experiments in embryos produced from unsorted semen, which results in a population of both male and female embryos. From a total of 300 zygotes, 90, 90 and 120 zygotes were subjected to no injection/control (Control), scramble siRNA injection (Scramble) and *TSPY* siRNA injection (TSPY KD), respectively. Sex ratio was determined at the blastocyst stage in each group, showing significant sex ratio shift towards female in the TSPY KD group. The sex ratios from the three groups are shown in Figure 5. Specifically, in the control group, out of 90 zygotes, 17 embryos reached the blastocyst stage (18.9%) of which 8 were males and 9 were females, resulting in male:female ratio = 1:1.25 ratio. In the Scramble group, out of 90 zygotes, 15 reached blastocyst stage (16.7%), of which 7 were males and 8 were females, resulting in a ratio male:female = 1:1.4. In the *TSPY* KD group, out of a total of 120 zygotes, 11 reached the blastocyst stage (9.2%) with 1 male and 10 females for a ratio of male:female = 1:10.

#### 2.5.2. Sex-Sorted Semen (X-Bearing)

Female zygotes produced from the three bulls in the knockdown experiment (Control, Scramble siRNA and *TSPY* siRNA) were analyzed for developmental parameters, as shown in Table 2 and Figure 6A,B. Cleavage rates of embryos produced from different bulls and derived from the different injection groups are shown in Figure 6A. There were no significant differences in the cleavage rate in different groups (Control, Scramble, TSPY KD). Similarly, the blastocyst rate did not show significant differences among bull groups or among different knockdown groups, as shown in Figure 6B.

#### 2.5.3. Sex-Sorted Semen (Y-Bearing)

The three different groups (Control, Scramble, *TSPY* KD) of male zygotes produced from Y-bearing semen of the three bulls were analyzed for developmental parameters, which are shown in Table 3. The cleavage rate of the *TSPY* KD groups was significantly decreased compared with the Control and Scramble groups in all the three bulls, as shown in Figure 6C. Specifically, the cleavage rates (%) were as follows: 1Y. 77.3 ± 3.5 (Control), 77.3 ± 2.9 (Scramble) and 39.6 ± 5.4 (*TSPY* KD); 2Y. 78.7 ± 4.5 (Control), 72.0 ± 4.0 (Scramble) and 41.4 ± 4.0 (*TSPY* KD); and 3Y. 80.7 ± 3.2 (Control), 72.0 ± 4.5 (Scramble) and 38.1 ± 3.1 (*TSPY* KD). Overall, only the *TSPY* KD group showed a significantly lower cleavage rate compared with the other injected groups. Blastocyst rates in the different knockdown groups were as follows: 1Y. Control 31.9 ± 1.8, Scramble 30.2 ± 1.9, *TSPY* KD 2.3 ± 1.2; 2Y. Control 32.2 ± 1.9, Scramble 30.6 ± 1.7, *TSPY* KD 2.6 ± 1.3; and 3Y. Control 40.5 ± 3.1, Scramble 38.9 ± 2.3, *TSPY* KD 2.5 ± 0.8. Overall, the blastocyst rate was significantly decreased in all *TSPY* KD groups, independently of the Y-bearing bull sperm used (Figure 6D).

#### 2.5.4. *TSPY* mRNA Expression

*TSPY* mRNA was measured in the three knockdown blastocyst groups produced from Y-bearing semen from the three bulls. As shown in Figure 6E, all the three *TSPY* KD groups exhibited no *TSPY* transcripts. In contrast, no significant differences were observed among the no-injection controls or among the three scramble injected groups, although all the scramble injected groups exhibited a significant decrease in *TSPY* transcript levels compared with the no-injection control groups.

## 3. Discussion

The Y chromosome has evolved to be shorter in size and have a low gene content, of which less than 5% is homologous to the X chromosome [33]. Containing more repeats in the ampliconic region has efficiently enabled the intrachromosomal non-allelic homologous recombination (NAHR), which results in copy number variation (CNV) [6]. *TSPY* shows CNV in a species-specific manner, among different individuals and within the same families [15,16,17,18,19]. The present study has demonstrated that (1) *TSPY* is uniquely expressed in male embryos, (2) *TSPY* CN and transcript levels were higher in male embryos with higher developmental capacity, and (3) knockdown of *TSPY* resulted in significant development failure of male early embryo development. Our results support that TSPY is needed for male development as the lack of it (in *TSPY* knockdown embryos) results in male embryos’ failure to develop to day 8 post fertilization in vitro. There is rising interest in studying CNV due to the growing evidence of a correlation between CN and diseases that have been reported in human studies. Fertility has also been correlated with the CN of specific Y-linked genes [34]. In cattle, CNV appears to correlate with fertility and semen parameters such as sperm concentration, quantity, motility and adjusted non-return rate [35]. Of those multi-copy Y-linked genes, *TSPY* has the largest CNV in bovines. *TSPY* CNV has been detected among breeds, bulls within the same breed, aging bulls and blastocysts [30]. The present study was successful in measuring *TSPY* copy number in individual male blastocysts and in correlating the presence/lack of TSPY with developmental capability in a sex-specific manner. As expected, there was no TSPY detected in the female blastocysts. Our results show not only *TSPY* CNV among brother embryos, but also among sperm from different donor bulls used (Appendix A). The average *TSPY* CN in sperm from bulls 1, 2 and 3 was 21, 22 and 42, respectively (Appendix A). The average *TSPY* CN in brother blastocysts from bulls 1, 2 and 3 was 30.2, 33.0 and 82.3, respectively (Figure 2D). According to a previous study in determining *TSPY* CNV in bulls, the differences between the highest and lowest *TSPY* CN were approximately 100 copies [19]. In male blastocysts, the difference between the highest and lowest *TSPY* CN was 52.1 *TSPY* copies. Unfortunately, we could not confirm *TSPY* CN or the fertility parameters of the three bulls that provided the sex-sorted semen for the present study. Determining the adult bulls *TSPY* CN would have provided more concrete assumptions on fertility parameters in males and would have supported the possibility of recombination events between pre-implantation embryos throughout the adult stages. De novo recombination has been observed in other studies. Oluwole et al., 2017 have shown de novo *TSPY* CN differences in somatic tissues as well as changes in *TSPY CN* of approximately 1~1.5 fold in 26% of aging bulls [22]. These changes in *TSPY* CN correspond to a doubling of average *TSPY* CN from sperm (21 copy variations) to male blastocysts (52.1 copy variations) and a further increase in adult bulls (100 copies variations). It is evident that recombination occurred after the fertilization, but the exact timing is unknown as we were not able to detect CN during the earlier embryo developmental stages due to technical detection limits and the minute DNA samples we were working with. *TSPY* mRNA levels were indeed detected throughout the early embryo developmental stages from the 2-cell to blastocyst stages. In female embryos there was no *TSPY* mRNA expression at any developmental stage as there is no *TSPY* gene in females. However, it should be noted that Li et al., 2008 have suggested that there is a sequence on the autosomes with homology to the *TSPY* that can be detected in females [36] but this is not supported by our or other previous studies [19,20,37]. In male embryos, *TSPY* mRNA expression followed the common embryonic genomic activation (EGA) profile, similar to an inverted logarithmic pattern, from the 2-cell to blastocyst stages, with a significant increase in *TSPY* transcripts at the 8-cell stage when the majority of the embryonic genome becomes activated in bovines. It has been shown that the Y-chromosome linked sex determinant gene *SRY* is expressed as early as the 4-cell stage in bovines [38] during the first minor wave of embryonic transcription before the 8-cell stage [39]. *TSPY* mRNA expression follows the same profile during this minor genomic activation. Genes expressed during this period are mainly housekeeping genes important for growth, development, metabolism and differentiation [40]. Caudle et al., 2013 showed that sex-specific genes on the Y-chromosome are among the first to be transcribed from the embryonic genome [30]. Similarly, our group has also shown that the female specific *XIST* gene, which initiates X-inactivation, is transcribed during the first wave of transcription [41]. This supports that sex-specific gene transcription is an important event for early embryonic development. In addition, this study shows higher *TSPY* CN and transcription to be associated with male embryo developmental competency. Embryos produced from the Y-bearing semen from bull 3 showed a higher percentage of blastocyst formation along with higher *TSPY* mRNA transcripts and higher *TSPY* CN. This suggests (1) the number of *TSPY* copies correlates with male embryo developmental capability and (2) the number of *TSPY* copies possibly enables more transcription to occur. There are conflicting reports on whether *TSPY* CN correlates with transcription level [20,42]. We have previously shown no correlation between *TSPY* CN and transcription level in adult bulls [19]. However, in this study, we found that embryos with the highest *TSPY* CN had more corresponding mRNA transcripts.

TSPY protein was observed only at the blastocyst stage in males, as shown in Figure 4. TSPY proteins were not detectable in the earlier stages or in the female blastocysts. TSPY is known to be involved in cell proliferation, which is an important process during the early embryo development. Expression of TSPY protein might be detectable only at the blastocyst stage, as *TSPY* transcription increases at the 8-cell and 16-cell stages to support further proliferation/development to the morula and blastocyst stages. A quick turnover and use of TSPY protein during the earlier developmental stages may explain why it was not detected in any amount until the blastocyst stage. TSPY function was investigated by microinjecting one-cell zygotes with small interfering RNA (siRNA) and monitored for developmental parameters, as well as *TSPY* transcription and translation. As shown in Figure 6, there was a lack of male embryo development after *TSPY* knockdown. It is possible that the knockdown procedure might affect embryo viability. To control for this, a knockdown of non-significant target scramble siRNA, as well as a non-microinjected control group were included in the experimental design. No differences in the scramble knockdown groups compared with the no-injection control groups were detected, supporting the specificity of *TSPY* knockdown and its effects on male development. An over 50% decrease in the percentage of cleavage in the *TSPY* knockdown embryos suggests a possible inhibition of de novo transcription or inhibition of the carried-over sperm transcripts. No *TSPY* mRNA or protein were detected in the female embryo groups, and *TSPY* knockdown did not show any effect in the female embryo groups; these results were expected as *TSPY* is a Y-linked gene. The presence of *TSPY* transcripts in the male embryos suggests that de novo transcripts are activated during EGA since the critical effect of *TSPY* knockdown in the male embryo is thought to be largely related to inhibition of the embryonic *TSPY* transcripts. This again supports *TSPY* de novo transcription at EGA. Gene transcription during or immediately following EGA is important for initiation and maintenance of early embryo development [43]. In more detail, during the minor genomic activation, genes are involved in repression of cell cycle and mitosis, development of tight junctions and compaction, and pluripotent stem cell maintenance and development. Genes activated during EGA have been shown to be involved in translation, ATP metabolic processes and RNA processing and differentiation [44]. In accordance with our results of a significant increase in *TSPY* transcripts at the 8-cell stage, Jiang et al., 2014 showed that genes involved in proliferation and differentiation were upregulated at EGA [45]. TSPY is known to be involved in proliferation and cell cycle progression during spermatogenesis [46]. The significant increase in *TSPY* at the 8-cell stage also supports its possible role in proliferation because development after the 8-cell stage becomes more rapid and embryos start to differentiate into diverse cell lineages [47]. Interestingly, spermatogenesis and early embryo development share some similarities with the rapid cellular division without cytoplasmic growth that is governed by various proliferation genes [48,49]. In addition, other Y-linked genes, such as *DDX3Y, USP9Y and ZRSR2Y*, have been shown to be expressed in the male bovine blastocysts before gonadal development [30]. In arrested male embryos, there was no expression of *USP9Y* and *ZFY* [30], which highly supports the contribution of the Y-linked genes in early embryo developmental potential. It is uncertain whether *TSPY* transcripts in the earlier stages of embryo development, before EGA, are from transcripts carried over to the zygote by the sperm or are the product from the minor first wave of embryonic genome transcription, as global Y-linked transcription studies in sperm and embryos were not performed. Surely the most novel and interesting data is that *TSPY* knockdown had a critical effect on male embryo survival. A previous knockdown study of another Y-linked gene, *USP9Y*, showed a reduction in the percentage of cleavage, but embryos that did cleave were able to develop into blastocysts [30]. This is in contrast with the results presented here, where *TSPY* knockdown resulted in almost complete lack of development to the blastocyst stage. As both genes are involved in spermatogenesis, it is clear that knockdown results strengthen the key role of TSPY during early embryo development. The results of this study strongly support that TSPY is required for the development of male, but not female, embryos. Some reasons for the developmental importance of TSPY in males might include: (1) Y-linked genes evolved to be involved in male survival and inheritance, and (2) *TSPY* might share similar molecular functions with autosomal genes. The Y chromosome has evolved from the ancestral autosomal gene, which resulted in a smaller size and gene content compared with the X chromosome and in the number of multi-copy Y-linked genes. Several studies have investigated the role of the Y-linked genes in male reproduction and development. At the molecular level, Y-linked genes are involved in many regulatory cell pathways not limited to the male reproductive system, but also in different organ systems. Similarly, *TSPY* has been shown to share similar functional properties with autosomal genes involved in proliferation and cell cycle regulation [25]. Male embryos tend to show increased expression of proliferation-related genes, which results in faster development and higher mitotic indices [50]. Thus, we can speculate that inhibition of *TSPY* results in early embryo arrest at the 8-cell stage. The present study is the first to correlate a Y-linked gene with early embryo developmental capability. We found that not only is *TSPY* important for male fertility but it also plays a crucial role in male embryo development and survival.

## 4. Materials and Methods

### 4.1. In Vitro Embryo Production (IVP)

Bovine embryos were produced in vitro following the IVP protocol as described previously (Rho et al., 2018) [39]. *Bos taurus* ovaries were obtained from a Canadian government-inspected abattoir (Cargill Meat Solution, Guelph, ON, Canada) in accordance with the Canadian Council on Animal Care and University of Guelph’s Animal Care Committee guidelines. Briefly, cumulus-oocyte complexes (COCs) were aspirated for in vitro maturation (IVM) in the IVM media (TCM 199) micro-drops containing 25 mM HEPES, 400 µL FBS, 1 µg/mL LH, 0.5 µg/mL FSH (Bioniche, Belleville, ON, Canada) and 1 µg/mL oestradiol (E2; Veterinary Chiron, Guelph, ON, Canada) and incubated at 38.5 °C with 5% CO_2_ for 20–24 h covered in mineral oil. A total of 4567 oocytes were collected and analyzed. Matured oocytes were in vitro fertilized (IVF) with sex-sorted cryopreserved semen from sexually mature Holstein bulls (Semex, Guelph, ON, Canada). Cryopreserved semen samples were prepared by Percoll gradient adjusted to approximately 10^6^ sperm/mL concentration, and 10–15 μL was spread onto IVF micro-drops containing IVF TALP supplemented with 1.94 mg caffeine/mL and 10 μg heparin at 38.5 °C with 5% CO_2_ for 18 h. Presumptive zygotes were then stripped to remove cumulus cells, washed and transferred into in vitro culture (IVC) micro-drops and incubated under mineral oil at 38.5 °C in 5% CO_2_, 5% O_2_ and 90% N_2_. IVP early embryo groups from unsorted and sex-sorted bull semen are shown in Table 4.

### 4.2. Microinjection

Microinjection was performed between 16 and 18 h post fertilization at the zygote stage, before transferring the zygotes to in vitro culture (IVC) media drops. Briefly, the zygotes were washed to remove excess sperm and granulosa cells and divided into 3 groups: no-injection control (Con); microinjection of custom-designed scramble siRNA (scram); and microinjection of *TSPY* siRNA (*TSPY* KD) (Invitrogen) (Table 5 and Table 6). Microinjection was performed on an inverted microscope (Leica DMIRE2) at 60X magnification with 150 hpa injection pressure, 15 hpa compensation pressure and 0.2 s injection duration for a volume of 15 pL [39,51], with fluorescence confirmation in the cytoplasm. The injected zygotes were transferred into IVC micro-drops for culturing as described above.

### 4.3. DNA Extraction and Digestion

Individual embryos underwent DNA extraction using forensicGEM^TM^ Universal (ZyGem NZ Ltd., Hamilton, New Zeland) DNA extraction kit, which uses thermophilic and mesophilic enzymes. Briefly, the embryos were placed in the forensicGEM universal mixture (1 uL of buffer, 0.1 uL forensicGEM and 1 uL of HISTOSOLV) and incubated in a thermal cycler at 52 °C for 5 min (cell wall degradation), 75 °C for 5 min (activation of proteinase for cell lyses) and 95 °C for 3 min (deactivation of proteinase). The extracted DNA was stored at −20 ℃ for further analysis. Prior to DNA copy number validation, the extracted DNA underwent digestion using a restriction enzyme to cut out single *TSPY* repeats from a tandem repeat from individually collected samples. As for the single repeat control, single allele, sex determining region (*SRY*) was used. BsoBI restriction enzyme (New England BioLabs^®^ Inc., ON, Canada) was selected as the cutting site does not interfere with the primer binding site of either *TSPY* or *SRY*. Briefly, 1.25 uL of 10X buffer and 0.25 uL BsoBI enzyme were added to the 10 uL of embryo DNA sample, incubated at 37 °C for 10 min and then immediately underwent digital droplet PCR (ddPCR) analysis.

### 4.4. RNA Extraction and Reverse Transcription

RNA was extracted using RNeasy^®^Plus Micro Kit (Qiagen, Toronto, ON, Canada) following the modified manufacturer’s guidelines, as previously described [52]. Briefly, 350 uL of buffer RLT Plus was added to each sample, vortexed for 30 s, transferred to the gDNA Eliminator spin column, then centrifuged for 30 s at 8000× *g*. Then, 70% ethanol was added to the flow-through, transferred to the Rneasy MinElute spin column and centrifuged for 15 s at 8000× *g*. Next, 80% ethanol was added to the Rneasy MinElute spin column and centrifuged for 2 min at 8000× *g*. Finally, 17 uL of Rnase-free water was added to the Rneasy MinElute spin column with a 1.5 ml collection tube attached and centrifuged at full speed for 1 min to elute RNA. The isolated RNA was immediately reverse transcribed to cDNA by adding 4 uL of qScript cDNA Supermix (Quantabio biosciences^TM^, MA, USA) with thermal incubation at 25 °C for 5 min, 42 °C for 30 min, 85 °C for 5 min, and finally stored at 4 °C. The reverse transcribed cDNA samples were stored at −20 °C until further analysis.

### 4.5. Digital Droplet PCR (ddPCR)

*TSPY* CN and expression was determined using digital droplet PCR. Briefly, ddPCR quantifies the target DNA and background DNA from the partition of the reaction mixes that are randomly distributed into 20,000 nL sized aqueous droplets using the QX200 droplet generator (Bio-Rad, ON, Canada). For *TSPY* CN analysis, multiplex PCR was performed using *TSPY* and *SRY primers* and corresponding TaqMan probes. The primers and probes with internal ZEN quencher were custom-designed, verified using IDT^®^ (Integrated DNA Technologies, Coralville, IA, USA), and are listed in Table 7. A modified reaction mix and thermal cycle PCR protocol were utilized. The 22 uL reaction mix, consisting of 6 uL of digested DNA sample, 10 uL ddPCR supermix for probes (no dUTP), 1.8 uL target F/R primer, 0.5 uL target probe, 1.8 uL reference F/R primer, 0.05 uL reference probe and 1.85 H_2_O, was added to a 96-well PCR plate. The final concentrations for each primer and probe were 900 nM and 250 nm, respectively. The PCR plates underwent droplet generation and were sealed for a 2-step thermal cycle reaction: 95–10 min, 45 cycles of (94 °C for 30 s, 60 °C for 60 s) and 98 °C for 10 min using Bio-Rad C1000 and finally stored at 4 °C. The ramp rate was set to 2 °C at all steps. The PCR plate was then transferred to the QX100 Droplet Reader (Bio-Rad) to analyze the individual droplets. Singleplex PCR was performed for *TSPY* expression analysis using *TSPY* and *YWHAZ* as a reference gene, as listed in Table 8. In a total reaction volume of 22 uL, 5 uL cDNA samples, 11 uL 2X Q200 ddPCR EvaGreen Supermix, 1.1 uL F/R primers (final concentration 250 nm) and 4.9 uL H_2_O were added to a 96-well PCR plate. The droplet generation, thermal cycle reaction protocol and droplet reading methods were as described above.

*TSPY* CN was determined as absolute copies/uL against single copy *SRY* gene. *TSPY* expression was determined as absolute copies/uL, and the reference gene *YWHAZ* was used as a quality control of the sample. Biological replicates of 30 single embryos and 15 pools of 50 sperm from each group underwent *TSPY* CN and transcript experiments.

The PCR reaction mix was partitioned into over 100,000 individual nano droplets. These individually partitioned droplets underwent amplification using a thermal-cycler and the end point of the reaction was inserted into the digital droplet reader (Bio-Rad). Due to independent readings being performed on each nanodroplet, the results reflect a more precise positive-negative call detection of the molecules.

### 4.6. Immunocytochemistry

TSPY protein was localized at different embryo stages by immunocytochemistry. Briefly, live embryos were washed 3X in 0.3% PBS/PVA, fixed in 4% Paraformaldehyde (PFA) at room temperature (RT) for 1 h, then stored in 2% PFA at 4 °C for staining. The embryos were blocked in the blocking solution (1XPBS + 0.01% Triton X100 + 5% Normal Donkey Serum (NDS)) for 1 h at RT, washed 3X in PBS/PVA and incubated with primary antibody (mouse anti-TSPY (H-11); sc-137050 Santa Cruz, TX, USA) in antibody dilution buffer (1:200 = TSPY:1XPBS + 0.005% Triton X100 + 0.05% NDS) overnight at 4 °C. The embryos were washed 3X in PBS/PVA for 10 min intervals at RT. From this step forward, the lights were reduced. The embryos were transferred to secondary antibody dilution buffer (1:200 = goat-anti-Mouse: 1XPBS + 0.005% Triton X100 + 0.05% NDS) for 2 h at 37 °C, then washed 1X in PBS/PVA followed by 30 min nucleus staining (1:2000 = Hoechest:1XPBS + 0.005% Triton X100 + 0.05% NDS) at 37 °C. Finally, the embryos were washed 3X in PBS/PVA for 10 min intervals and single embryos were transferred onto a slide with 1 uL of vector shield drop, covered with cover slip and sealed. The slides were stored at 4 °C in the dark until confocal microscope analysis.

### 4.7. Confocal Imaging and Quantification

The immunofluorescent stained embryo and sperm samples were visualized under the Olympus Fluoview Laser Scanning Confocal System on an IX81 inverted microscope and the images were captured under 40X and 60X magnification for the embryos and sperm, respectively. All appropriate controls were applied: HV laser intensity was adjusted below the threshold for each dye and the laser adjustments were not changed throughout the imaging. The images were captured in TIFF format and relative quantification of TSPY protein was performed using image J software v1.53 (https://imagej.nih.gov), with intensity of TSPY fluorescent and background measured 3X. These measurements were calculated using the corrected total cell fluorescence (CTCF) equation which is CTCF = Integrated density − (Area of selected cell × Mean fluorescence of background readings).

### 4.8. Statistics

The cleavage and blastocyst rates were analyzed as cleavage against total number of oocytes, and blastocysts over the total number of cleaved embryos using the Wilcoxon rank test and Fisher exact test in the Statistical Analysis System (SAS) program. All values are expressed as means ± SEM, with significant differences considered as *p*-value < 0.05.

The knockdown results were analyzed by linear regression analysis and the correlation between blastocyst rate and sex ratio by simple linear regression (Prism GraphPad). Additionally, the odds ratio was calculated to determine whether the sex ratio in each treatment group differs significantly from the expected 1:1 ratio (MedCalc statistic software MedCalc Version 20.216). Significant differences were considered as *p*-value < 0.05.

*TSPY* CNV in brother blastocysts was analyzed by Prism GraphPad software (GraphPad Prism 9.4.1.681). Differences in the *TSPY* CN were compared using one-way ANOVA and Tukey’s multiple comparison test. All values are expressed in ± SEM and significant difference was considered as *p* value < 0.05.

TSPY expression and protein level were analyzed with one-way analysis of variance (ANOVA). Tukey’s multiple comparison tests were also performed to evaluate differences between specific cell stages of specific bulls or differences between the bulls at specific cell stages using graph pad prism. Differences in expression levels between bulls 1, 2 and 3 at each cell stage were analyzed using unpaired Student’s *t*-test (graph pad prism). Significant differences were considered as *p* < 0.05.

## Figures and Tables

**Figure 1 ijms-24-03349-f001:**
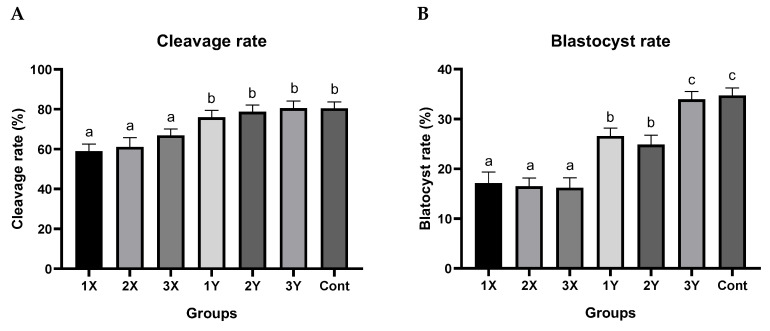
Developmental percentage of early embryos produced from unsorted and sex-sorted (X- and Y-bearing) semen from bulls 1, 2 and 3. (**A**) represents the percentage of cleavage in each female (1X, 2X, 3X), male (1Y, 2Y, 3Y) and control groups. (**B**) represents the percentage of the blastocyst in each female (1X, 2X, 3X), male (1Y, 2Y, 3Y) and control groups. Each group were includes 10 biological replicates in which a total of 2557 oocyte were used. The data is shown as mean ± SEM, different superscript letters above the columns indicate significant differences within each group (*p* < 0.05).

**Figure 2 ijms-24-03349-f002:**
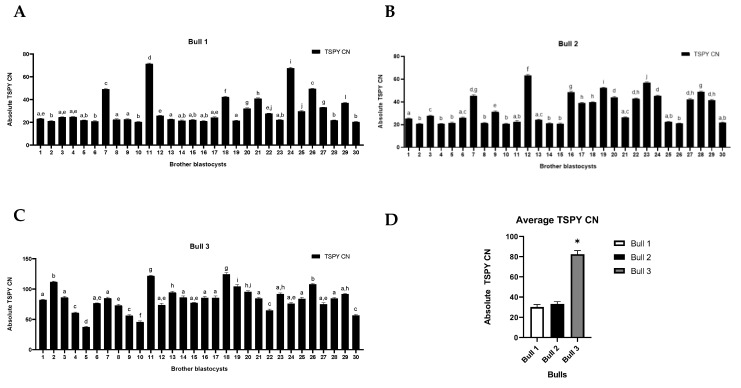
*TSPY* CNV in male brother blastocysts. (**A**–**C**) show *TSPY* CN in brother blastocysts from bull 1, 2 and 3, respectively. Experiments were conducted with 30 biological replicates. (**D**) represents the average *TSPY* CN from brother blastocysts. The data are represented as mean ± SE; different superscripts above the bars in each graph indicate significant differences (*p* < 0.05) and * indicates significant differences (*p* < 0.05).

**Figure 3 ijms-24-03349-f003:**
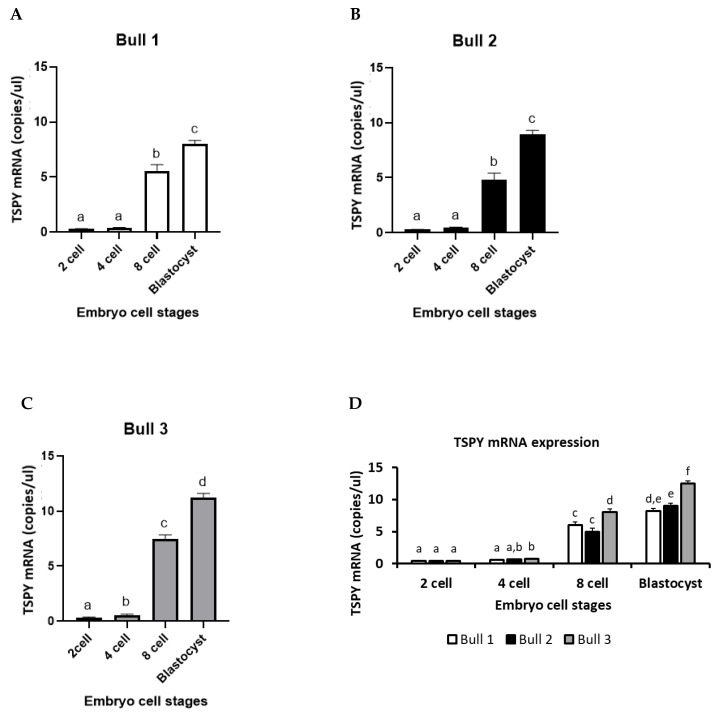
*TSPY* transcripts during the early embryo developmental stages in male groups. (**A**–**C**) show *TSPY* mRNA transcripts at 2, 4, 8 cells and blastocyst in male embryos produced from Y-bearing semen from bulls 1, 2 and 3. Experiments were conducted in 30 biological replicates for each cell stage. *TSPY* was quantified by absolute copies/uL using ddPCR. The data is shown as mean ± SEM; different superscript letters above the columns in each graph indicate significant differences between the cell stages (*p* < 0.05). (**D**) represents *TSPY* mRNA transcripts of all male embryo groups together.

**Figure 4 ijms-24-03349-f004:**
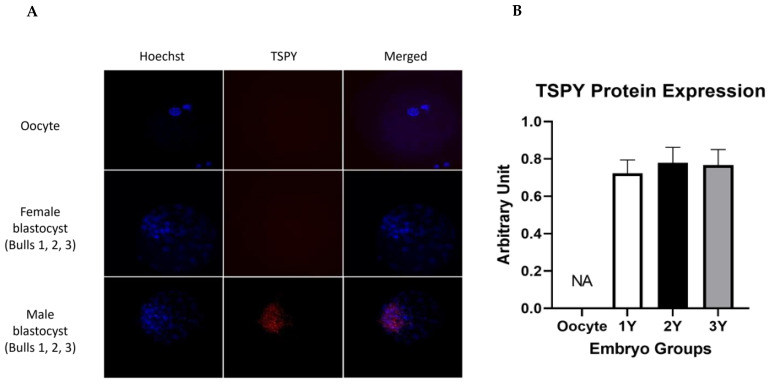
TSPY protein localization and expression in the blastocyst. The image in (**A**) represents TSPY protein localization in the oocyte, female and male blastocysts. Hoechst represents the nucleus in blue, TSPY represents TSPY protein in red and Merged represent a combined image of both nuclear and TSPY protein staining. Olympus Fluoview Laser Scanning Confocal System on IX81 inverted microscope was used and images were captured at 60X magnification. (**B**) represents the TSPY protein quantification corresponding to the images for the oocyte and male blastocyst. Experiments were conducted in 15 biological replicates and 5 technical replicates. The data is shown as mean ± SEM.

**Figure 5 ijms-24-03349-f005:**
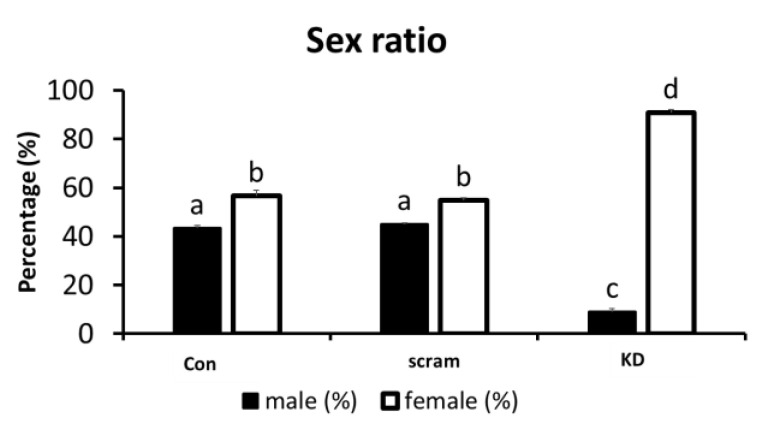
Sex ratios in different knockdown groups in embryos from unsorted semen. Groups are represented as follows: con, no-injection control group; scram, scramble siRNA injection group; and KD, *TSPY* siRNA injection groups. The data is shown as mean ± SEM for 5 biological replicates; different superscript letters indicate significant differences (*p* < 0.05).

**Figure 6 ijms-24-03349-f006:**
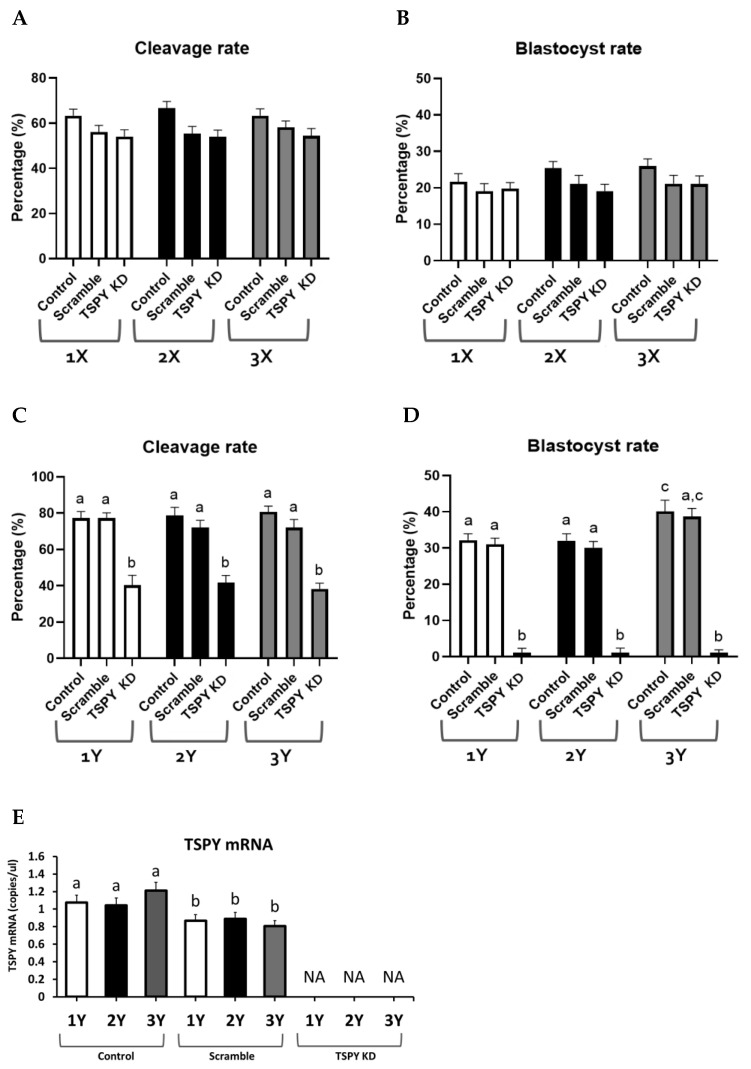
Developmental percentage in male embryo groups in knockdown. (**A**,**B**) represent the cleavage and blastocyst percentage in female embryo groups (1X, 2X, 3X) produced from X-bearing semen, respectively. (**C**,**D**) represent the cleavage and blastocyst percentage in male embryo groups (1Y, 2Y, 3Y) produced from Y-bearing semen, respectively. (**E**) represents *TSPY* mRNA expression in male embryo groups. Groups are represented as: Control, no-injection; Scramble, scramble siRNA injection; and *TSPY* KD, *TSPY* siRNA injection. The data is shown as mean ± SEM. Each group included 5, 15 and 25 biological replicates and 2 technical replicates for Control, Scramble and *TSPY* KD, respectively, where 1 biological replicate consists of a pool of 30 oocytes. Different superscript letters represent significant differences (*p* < 0.05).

**Table 1 ijms-24-03349-t001:** Developmental percentage and total cell number from IVP embryos produced from unsorted and sex-sorted bull semen.

Groups	Oocytes	Cleavage		Blastocysts		Blastocyst Cell Number	
	No.	No.	Percentage (%)	No.	Percentage (%)	No.	Mean ± SEM
1X	356	210	58.9 ± 3.6 ^a^	36	17.1 ± 2.2 ^a^	15	115.4 ± 15.1
2X	298	182	61.0 ± 4.7 ^a^	30	16.5 ± 1.7 ^a^	15	121.1 ± 12.4
3X	314	210	66.9 ± 3.2 ^a^	34	16.2 ± 2.0 ^a^	15	124.7 ± 9.3
1Y	416	316	75.9 ± 3.4 ^b^	84	26.6 ± 1.6 ^b^	15	120.7 ± 14.0
2Y	460	362	78.6 ± 3.4 ^b^	90	24.9 ± 1.9 ^b^	15	116.1 ± 10.5
3Y	534	430	80.5 ± 3.5 ^b^	146	34.0 ± 1.6 ^c^	15	119.6 ± 7.4
Con	179	144	80.4 ± 3.2 ^b^	50	34.7 ± 1.6 ^c^	15	122.3 ± 8.6

Values in the same column with different superscript letters indicate significant differences (*p* < 0.05).

**Table 2 ijms-24-03349-t002:** Female embryo developmental parameters in *TSPY* knockdown groups.

Groups		Oocytes	Cleavage		Blastocyst		
Knockdown	Bulls (X)	No.	No.	%	No.	% from No. Oocytes	% from No. Cleavage
Control	1(X)	150	94	63.3 ± 2.9	19	12.7 ± 2.7	21.7 ± 2.2
	2(X)	150	100	66.7 ± 3.0	25	16.7 ± 2.3	25.4 ± 1.9
	3(X)	150	94	63.3 ± 3.1	25	16.7 ± 2.4	26.6 ± 2.0
Scramble	1(X)	150	85	56.1 ± 2.9	18	12.0 ± 2.5	21.2 ± 2.0
	2(X)	150	82	55.5 ± 3.1	15	10.0 ± 2.8	18.3 ± 2.3
	3(X)	150	87	58.1 ± 2.9	18	12.0 ± 2.8	20.7 ± 2.3
TSPY KD	1(X)	150	80	54.1 ± 3.0	15	10.0 ± 2.0	18.8 ± 1.6
	2(X)	150	80	53.9 ± 3.1	15	10.0 ± 2.3	18.8 ± 1.9
	3(X)	150	83	54.4 ± 3.2	18	12.0 ± 2.8	21.7 ± 2.3

**Table 3 ijms-24-03349-t003:** Male embryo developmental parameters in TSPY knockdown groups.

Groups		Oocytes	Cleavage		Blastocyst		
Knockdown	Bulls (Y)	No.	No.	%	No.	% from No. Oocyte	% from No. Cleavage
Control	1(Y)	150	116	77.3 ± 3.5 ^a^	37	24.7 ± 2.0 ^a^	31.9 ± 1.8 ^a^
	2(Y)	150	118	78.7 ± 4.5 ^a^	38	25.3 ± 2.1 ^a^	32.2 ± 1.9 ^a^
	3(Y)	150	121	80.7 ± 3.2 ^a^	49	32.7 ± 3.4 ^c^	40.5 ± 3.1 ^c^
Scramble	1(Y)	450	346	77.3 ± 2.9 ^a^	104	23.1 ± 1.9 ^a^	30.2 ± 1.9 ^a^
	2(Y)	450	324	72.0 ± 4.0 ^a^	99	22.0 ± 1.9 ^a^	30.6 ± 1.7 ^a^
	3(Y)	450	324	72.0 ± 4.5 ^a^	126	28.0 ± 2.5 ^a,c^	38.9 ± 2.3 ^a,c^
TSPY KD	1(Y)	750	300	39.6 ± 5.4 ^b^	7	0.9 ± 1.4 ^b^	2.3 ± 1.2 ^b^
	2(Y)	750	307	41.4 ± 4.0 ^b^	8	1.1 ± 1.4 ^b^	2.6 ± 1.3 ^b^
	3(Y)	750	285	38.1 ± 3.1 ^b^	7	0.9 ± 0.9 ^b^	2.5 ± 0.8 ^b^

Values in the same column with different superscript letters indicate significant differences (*p* < 0.05).

**Table 4 ijms-24-03349-t004:** In vitro produced (IVP) early embryo groups from unsorted and sex-sorted bull semen.

Bulls	Sex Sorted	IVP Embryo Groups
Control	Unsorted (X-/Y-bearing sperm)	Con: Female/Male embryos
Bull 1	X-bearing sperm	1X: Female embryo
	Y-bearing sperm	1Y: Male embryo
Bull 2	X-bearing sperm	2X: Female embryo
	Y-bearing sperm	2Y: Male embryo
Bull 3	X-bearing sperm	3X: Female embryo
	Y-bearing sperm	3Y: Male embryo

**Table 5 ijms-24-03349-t005:** IVP embryo groups with knockdown treatments.

Knockdown	Bull Semen	IVP Embryo Groups
No injection (NI)	Bull 1	1(X)
		1(Y)
	Bull 2	2(X)
		2(Y)
	Bull 3	3(X)
		3(Y)
Scramble siRNA injection (Scram)	Bull 1	1(X)
		1(Y)
	Bull 2	2(X)
		2(Y)
	Bull 3	3(X)
		3(Y)
*TSPY* siRNA injection (TSPY KD)	Bull 1	1(X)
		1(Y)
	Bull 2	2(X)
		2(Y)
	Bull 3	3(X)
		3(Y)

**Table 6 ijms-24-03349-t006:** List of sequence specific siRNA used for microinjection.

siRNA	Sequence (5′-3′)	Primer Length	5′ Mods	3′ Mods	Genbank
*Scramble*	F-GGGCAUUUGGACUUCUCAU	21	Flu	-	NM_001244608.1
	R-AUGAGAAGUCCAAAUGCCC	21	Flu	-	
*TSPY*	F-GGGCUUACGUUCAGUUCAU	21	Flu	-	NM_001244608.1
	R-AUGAACUGAACGUAAGCCC	21	Flu	-	

Flu: Fluorescent, Mod: Modification.

**Table 7 ijms-24-03349-t007:** List of primer and probe sequences for genomic DNA analysis.

Name	Sequences	Length (N)	5′ Mod	Int	3′ Mod	Genbank
*TSPY Primer*	F-TGCTTCGAGGAAGACATCG R-CCTCCTCTGATGGTTCTTGC	19 20	- -	- -	- -	X74028.1
*TSPY Probe*	ACGCAGGGC/ZEN/TTACGTTCAGTTCAT	24	6-FAM^TM^	Zen^TM^	Lowa Black^®^ FQ	
*SRY Primer*	F-CCAATTAAGCCGGTCACAGT R-GCACAAAGTCCAGGCTC	20 20	- -	- -	- -	Z30327.1
*SRY Probe*	TCGGCGGAC/ZEN/TTTCCCTGTAACAAA	24	HEX^TM^	Zen^TM^	Lowa Black^®^ FQ	

Mod: Modification, Int: Internal, FAM: Fluorescein amidite, HEX: Hexachloro-fluorescein.

**Table 8 ijms-24-03349-t008:** Primer list.

Name	Sequences (5′-3′)	Product Size (bp)	Genbank	References
*TSPY*	F-CCCAGAATCGAACAGGATTG R-TTGTCTCTCACGGACGAACC	215	NM_001244608.1	Hamilton et al., 2012 [31]
*SRY*	F-CCAATTAAGCCGGTCACAGT R-ACAAGAAAGTCCAGGCTC	162	NM_001014385.1	Hamilton et al., 2012 [31]
*YWHAZ*	F-GCATCCCACAGACTATTTCC R-GCAAAGACAATGACAGACCA	102	NM_174814.2	Goossens et al., 2005 [53]

## Data Availability

The data underlying this article will be shared on reasonable request to the corresponding author.

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
