# Peer review of "Testis-Specific Protein Y-Encoded (TSPY) Is Required for Male Early Embryo Development in Bos taurus"

_ijms, 2023, doi:10.3390/ijms24043349_

Round 1
Reviewer 1 Report
This manuscript is very interesting and brings new and important information to this research field. It is a well-written and reports the association between the copy number variation of the TSPY gene and male embryo development using the bovine model. The methodology is adequate and results are well presented and discussed.
Although, some minor changes are needed to be clarified in order to improve the quality of this study for publication.
1. Please elaborate the data regarding the chosen sorted semen used in the study. Such as the age of the bulls; semen and some cellular features of the sperm of each bull (i.e., concentration, motility...an extra); the breed of the chosen semen; how does the semen was prepared for in vitro fertilization.
2. With respect to bull no. 3, a higher expression of TSPY mRNA has been reported in the blastocyst, however, no difference was found in the protein level. In parallel, higher blastocyst yield has been found in this group. Considering variation among bulls, it would be interesting if the author could addresses this point and add it to the discussion.
3. Please specify the samples collection procedure prior to their analysis. How many embryos from each stage has been analyzed (2-, 4- and 8-cells).
4. With respect to the blastocyst quality, does other parameters have been evaluated? such as degree of expansion, survival rate after cryopresevation, the evident of the ICM, appearance of fragmentation.
5. Some minor corrections:
- please add the abbreviation of IVC in line 366 instead of line 372.
- no SEM is presented in tables 2 and 3 in the %blastocyst out of total oocytes/ per replicate
Author Response
Please see pdf attached

Reviewer 2 Report
The article intitled Testis Specific Protein Y-encoded (TSPY) is required for male early embryo development in Bos Taurus, aimed to assess TSPY gene functions during early male development. For that, authors performed a series of experiments using different generated embryos of Bos taurus (which should be written with “t” and not with “T”). Experiments were, firstly the IVF with sex-sorted and not sorted semen. Then, quantitative embryo competency, evaluation of TSPY copy number, transcripts, and protein levels. Moreover, the authors were able to knockdown TSPY through siRNA in a very simple and elegant way. All the results have answered the questions brought in by the author and, for that reason, I indicate the article for publication at the International Journal of Molecular Sciences. Nevertheless, I have some minor and easy to do suggestions for the authors. Those are following bellow:
1. Explain a bit more about the TSPY in the abstract in the sentence bellow
Line 11: TSPY is a highly conserved multi-copy gene with copy number variation (CNV). I’d suggest something like “TSPY is a highly conserved multi-copy gene usually expressed on… (organisms, cells, and situation / moment), with copy number variation (CNV).
2. Include "to"
Line 37: “…, leading to germ cell loss which further results in infertility.”
3. Replace "with"by "and"
Line 47: “…Human has 30-60 13 TSPY CN, with and the most range of variation 47 shown in cattle with a range of 20-200 copies.”
4. This sentence is confusing and bellow I explain why:
Line 58: “… and plays an important role for promoting differentiation and proliferation in the biological machineries during male meiotic divisions, including spermatogenesis, spermatogonia cell renewal and prophase I spermatocyte differentiation”.
Male meiotic divisions are part of the process we call spermatogenesis. In the sentence above the authors present the opposite idea, i.e., the spermatogenesis is part of the phenomenon “male meiotic divisions”. Spermatogenesis encompasses a mitotic/self-renewal division stage, named proliferation or spermatogonial phase, the meiotic divisions, named spermatocyte or meiotic phase, and the spermiogenic phase, when spermatozoa maturate. So, I strongly suggest this sentence to be remodeled. For example: “… and plays an important role for promoting differentiation and proliferation during spermatogenesis, e.g., during spermatogonia proliferation / self-renewal, and at prophase I during spermatocytes differentiation.”
5. The authors use P<0.05 and p<0.05 along the text and figures legends. Please stablish a pattern, either P or p.
6. There is a difference in pixels between figures 2A and C compared with figure 2B. That may cause problems for readers of printed versions. If it’s possible, please, correct that.
7. In figure 2, put (D) in bold in the legend text.
8. In figure 4 the authors showed an IF for TSPY image together with a graphic of quantification of TSPY protein. Since there is no expression on oocytes and 1X, 2X and 3X, I suggest editing the graphic in a way there is only the quantification for the Y embryos, keeping the information about X in the text. In the way the graphic is now, it is not giving any extra information from what is written and occupying a space that could be used to increase the size of IF figure. Readers appreciate when IF figures come as big as possible.
9. In the Head of Tables 2 and 3 “Blastocyt” should be on bold.
10. Please, cite references for some information in Discussion section:
Line 215: Y chromosome has evolved in being shorter in size and with low gene content of which less than 5% is homologous to X chromosome.
Line 216: Containing more repeats in the ampliconic region has efficiently enabled the intrachromosomal non-alleleic homologous recombination (NAHR), which results in copy number variation (CNV).
Line 218: TSPY shows CNV in a species-specific manner, among different individuals and within the same families. Reference for this information has been cited in previous parts of the main text.
Line 305: Gene transcription during or immediately following the EGA is important for initiation and maintenance of early embryo development. In more details, during the minor genomic activation, genes are involved in repression of cell cycle and mitosis, development of tight junctions and compaction, and pluripotent stem cell maintenance and development. Genes activated during the EGA are shown to be involved in translation, ATP metabolic processes and RNA processing and differentiation. This long fragment of information lacks references.
Line 317: Interestingly, spermatogenesis and early embryo development share some similarities with the rapid cellular division without cytoplasmic growth, governed by various proliferation genes.
11. Put a full stop before “The”
Line 231: “…breeds, bulls within the same breed, aging bulls and blastocysts 22 The…”
12.
Line 258: “...but this is not supported by our or others’ previous 258 studies 28.” Here the authors mention others’ previous studies, in the plural. However, only one reference was cited.
13.
Line 246: “...De novo recombination has been observed in other studies. Oluwole et al., 2017 have shown de novo TSPY …”. This reference is cited differently from the pattern used so far in the article, with the date of publication (e.g., Lines 256 and 267). That happens all along the discussion section. Please check the pattern of citations stablished by IJMS.
14. When checking the editing of the article, I noticed that all the references in the main text and on the list are slightly different from the Microsoft World Template supplied by IJMS. Please check with the editor the need of changing to the template format pattern.
Author Response
Please see pdf attached
